# "Festival y Protesta": The Integral Role of Protesting State Violence in Celebrating Puerto Rican Women and Feminists

**Amaury J. Rijo Sánchez**

Department of Sociology, University of Illinois at Urbana-Champaign, Champaign, IL 61801, USA; amaury2@illinois.edu

**Abstract:** Eradicating the mistreatment of Puerto Rican women and people that local and U.S. governments enact has been a major transformative goal for Puerto Rican feminist movement communities. The celebration of International Working Women's Day presents optimum opportunities for organizations to celebrate and make visible the monumental achievements of Puerto Rican women and people. Similarly, they foster the opportunity to strategically protest the large-scale and harmful attacks of the United States and Puerto Rico's (abbreviated throughout as U.S. and P.R.) governing double-bind onto minority Puerto Rican populations. Feminist activists, protesters, artists, and attendees collaborate in performances, speeches, and overall programming, resulting in dually celebratory and protest-based marches. Further, the multifaceted approach to protesting observed at the celebration of International Working Women's Day shines light on decolonial and feminist efforts that bring about social justice and transformation. This article analyses ethnographic data collected through participant observation in one march held in Puerto Rico, as well as a small archive of news articles relating to said march. Results reflect strategic forms of organizing and protesting that exercise activists' agency in communication with the government and state. Further, they show the demand for accountability and action in favor of minority Puerto Rican populations. Simultaneously, the results also shine a light on the synergistic character of state and government approaches to minimize the impact of activist protesting.

**Keywords:** state violence; political queerness; social movement communities; Puerto Rican feminisms





## 1. Introduction

At the intersections of gender, race, class, age, and sexuality, Puerto Rican minorities find themselves in precariously vulnerable positions vis-à-vis the repressive character of state and governmental institutions in Puerto Rico [1–3]. The asphyxiating realities these institutions produce and reproduce result in displacement, socioeconomic precarity, and multiple forms of violence against women, LGBTQ+ or queer people, and low-income communities. The eradication of these multiple forms of violence that the state enacts is a major and transformative goal for feminist movement communities in the Puerto Rican archipelago [3,4]. While gender violence has been a continuous problem in Puerto Rico, its resistance and opposition have expanded and diversified in recent decades. Thus, strategic mobilization, organization, and institutionalization have characterized Puerto Rican feminist movement communities in great part today. In this article, I join the claims of other Puerto Rican scholars regarding the insidious, divisive, veiling, and violent character of Puerto Rican state and government structures and institutions. Further, I analyze how the cyclic disposition of celebration rooted in the communal and cultural character of Puerto Rican feminist movements presents a critical space for feminist movement communities to amplify the urgency of their demands in protest while commemorating the historical pertinence of protests themselves. Ultimately, I argue that the processes and tools feminist movement communities deploy in protest through celebratory sociopolitical events fuel resiliency in the face of the lived implications of activist organizing and contentious interactions with the state.

### 1.1. Puerto Rico: State and Government

Initially a Spanish colony and later under the continuous colonial rule of the United States, many argue Puerto Rico never stopped being a colony [3,5,6]. Under North American rule, Puerto Rico has been both an accomplice and victim to atrocious abuse, oversights, and experimentation at the hands of the United States government over decades of subjugation [7,8]. It has additionally been neglected and circumscribed to its erosive fate in the face of continuous hurricanes, earthquakes, swarms, and aftermath [9–11]. Certainly, not all the citizens of this Caribbean archipelago support the infrahuman circumstances these governing structures perpetuate; however, reaching common grounds on the best ways to move forward and garner the governmental support needed is complex and near impossible.

Puerto Rico's long colonial history has been characterized by economic precarity and institutional stress [1,6,12]. The latter half of the 20th century sedimented the interpenetration of the U.S. and P.R. state, legal, and political structures that maintained Puerto Rico under colonial subjugation. Further, the bipartisan rule of Puerto Rican political parties, namely the Popular Democratic Party (PPD) and the New Progressive Party (PNP), both support at different levels a relationship with The United States. These relationships and structures directly contribute to the solidification of Puerto Rico's geopolitical status while scaffolding a façade of Puerto Rico's legal and political autonomy.

In recent years, scholars have pinpointed and outlined the different forms of institutional control that scaffold Puerto Rico's colonial status [1,6,13,14]. These scholars stress how the repression and criminalization of socioenvironmental justice movements, as well as the cruel and stagnating logic of empire, contribute to the maintenance of Puerto Rico's coloniality. Austerity measures and policies have asphyxiated public resources in the name of the capitalist privatization and neo-liberalization of this Caribbean archipelago. Its results are reflected not only in the detrimental rates of unemployment but also in the endangerment of environmental resources and the reduction or mismanagement of public funds for labor and educational organizations and purposes.

Recent evidence of the United States's oppressive power over Puerto Rico is the implementation of the U.S. federal law Puerto Rico Oversight, Management, and Economic Stability Act (PROMESA) which established an unelected Financial Oversight and Management Board (colloquially addressed as *La Junta*) appointed by the U.S. to oversee the islands' finances and settle the Puerto Rican debt crisis. Specifically, this board is responsible for making decisions that prioritize restructuring an unpayable and unjust debt over the dignified livelihoods of Puerto Rican populations [9,10]. *La Junta's* administrative ruling enacts a colonial logic wherein the privatization of public services often results in subpar or depleted public resources and the exponential growth of the archipelago's cost of living. For example, the privatization of the archipelago's power grid has supported substandard electrical power systems [13]. Asphyxiating results that also support capitalizing from privatized services can be observed across other public institutions such as healthcare, housing, and education. Thus, settling the Puerto Rican debt is dehumanizing Puerto Rican populations. While many oppose Puerto Rico's subjugation to the United States, support for this subjugation is insidiously scaffolded through the economic and political interests of its local government and the few elite individuals that benefit from its fraudulent management.

Many scholars and activists argue that the enforcement of U.S. and P.R. state rule in Puerto Rico is notoriously violent, neoliberal, criminalizing, and punitive [1]. In the face of social and environmental justice activism, U.S. and P.R. state rule continues to respond with repression, organized austerity measures, criminalization of protest and activism, minority role manipulation in the service of police brutality, and disproportionate use of surveillance, among other oppressive tactics [2,4]. Alternatively, in the face of socioeconomic precarity and interpersonal violence, they continuously respond with neglect, abandonment, and politically inflamed promises that rarely have any societal impact nor reach material fruition [9]. These bad-faith politics are overwhelmingly distorted to dehumanize

minorities and garner the support of members of the middle and upper class in service of the governing few.

Deriving from the Du Boisian theory of double consciousness, Negrón-Muntaner develops the concept of political queerness to explain the political conditions that foster the coexistence of different ideological stands on Puerto Rico's political status in relation to the United States [6]. She argues that Puerto Rico's political queerness characterizes the global, national, and colonial discourses and positionalities that produce both subjects to colonial logic experienced in the islands and metropolitan citizens in the context of a broadly understood U.S. population. Alternatively, Lugo-Lugo makes use of this theory in literary analysis as a tool to challenge colonialism [15]. While Puerto Ricans have been subjugated to colonizing global powers for centuries, they maintain a vibrant command over their identities as more than just colonial subjects. In so doing, many Puerto Rican scholars and activists reaffirm their Puerto Rican identities in their search for and development of other ways of living and knowing that support their dignified lives and that of their marginalized compatriots. This politically queer terrain supports and perpetuates an overarching climate of frictions and contradictions through which sociopolitical transformations are both sped up and halted at different points in time. Simply put, the conditions that political queerness affords can be used to advance the sociopolitical transformations that activists pursue. Nevertheless, these conditions can also be used to the advantage of U.S. and P.R. governance and its efforts to defer or dismiss these advances.

*1.2. Puerto Rico: By the Women, for the People*

The overdetermining pressure of U.S. colonial imposition fostered multiple forms of resistance that have resiliently withstood imperial abuse. While the governing institutions that control Puerto Rican society show little concern for the dignified livelihoods of its populations, the ever-growing presence of leftists, feminists, community organizers, and grassroots activists in the public sphere resists and merits recognition [3,16,17]. These mobilizing agents conceive and support other ways of existing and knowing that support Puerto Rican livelihoods based on safety, dignity, and flourishment. Moreover, Puerto Rican women and feminists have, in many ways, spearheaded movements resisting the repressive character of U.S. and P.R. governance.

The entrance of women into the public arena ushered the proliferation of feminist movements in many places of Latin America and the Caribbean as well as in different parts of the globe. While transnational, mutual financial aid was not always accessible, collaboration with political power undeniably supported the emergence of feminist movements across the world [18]. The last thirty years of the 20th century evidenced this with the emergence and institutionalization of women-led labor unions, grassroots collectives, and non-governmental organizations (NGOs) rooted in feminist principles of social justice and equality [19,20]. Thus, we must understand feminist movements today in their heterogeneous, poly-centric, and expansive characterizations. Collaboration with political power for financial support often involved strategic alignment to traditional ideologies that not many feminists were keen on, thus contributing to the diversification of Puerto Rican feminist movements [18]. Particularly in the Puerto Rican archipelago, feminist organizers garnered political alliances and witnessed institutional changes earlier than in other places in Latin America. These processes are also fostered through Puerto Rican feminists' use of Puerto Rico's politically queer terrain. From political and community-based collaborations (both local and abroad), feminist movement communities grew, diversified, and have been institutionalized in recent times. Thereon, Puerto Rican feminist theories learn from and collaborate across national boundaries with keen attention to critical frameworks of globalization-from-below, Black feminist studies, as well as Decolonial, Latin American, and U.S. feminisms [12,19,21].

At the crossroads between North and Latin America, Puerto Ricans have not only circumvented knowledge and efforts in several contexts but have been at the forefront of many feminist trends both in and outside Puerto Rico [22]. Puerto Rican feminist scholars

and activists militantly denounce the sexist, racist, and elitist mistreatment of minority populations on multiple platforms and at multiple scales. They have also denounced the feminization of poverty and the erosive and detrimental realities that systemic conditions perpetuate. They expose the multilayered character of U.S. and P.R. governance and strategize to simultaneously bear witness, resist, raise awareness, and educate the masses against gender violence, capitalist evisceration, mass displacement, and neoliberal coloniality. A broad range of social actors have collaborated over the years in the development of feminist movement communities. I make use of social movement communities as a concept that includes all social actors who share and further the goals of a social movement to understand the breadth of Puerto Rican feminist movements [23]. These are individual movement adherents, organizations, alternative institutions, institutionalized movement supporters, and other sociocultural groups. Further, these members stand in solidarity with one another through the impact and commemoration of the historical struggles of feminist pioneers and are militantly devoted to decolonial pursuits represented under the concept of *vidas dignas*, or dignified lives [24,25]. Moreover, the concept of *vidas dignas* prioritizes horizontal relationships with nature and others. It stakes the claim that human life is worthy of having their needs met and their life valued by governing institutions. It is a framework that emerges from a decolonial emphasis on recuperation and regeneration. The concept of dignified lives has gained mainstream recognition through social movements in Latin America, and it is a point of departure for many engaged in feminist and social justice organizing in Puerto Rico [24].

Protesting the violence that governing structures in Puerto Rico enact has presented optimum grounds for building solidarity-based networks and strategic organizing among Puerto Rican minorities. Protests in Puerto Rico are often multisectoral and multifaceted, with cultural and artistic performances, public speeches, and peaceful manifestations of resistance. It provides protesters the opportunity to deploy their agency in aligning with modes of self-identification alternative to that of the dominant and governing structures. Further, it fosters a sense of cultural belonging that U.S. and P.R. governance often negates. While many protests exemplify this claim (e.g., University student protests, U.S. Navy settlement protests, and labor union protests), some of the most notable protests in recent times were the 2019 Summer Protests demanding the then-governor Ricardo Rosselló's resignation [5]. Ignited by the accidental exposure of a notorious group chat among government officials, including Rosselló, the veil was dropped between individual speculations about governmental neglect and the wickedly sexist, homophobic, and necropolitical iterations of government officials regarding Puerto Rican women and minorities.

Feminist movements spearheaded the protests demanding Rosselló's resignation, calling upon the public's *coraje* [3]. LeBrón's theory of *coraje* uses its Spanish definition, encompassing both rage and courage, to conceive of an activist tool to critically mobilize the population's agency beyond its colonial and capitalist obstruction. It challenges the limits of a Fanonian understanding of rage in colonial contexts to bring focus to the productivity of its affective register [3,26]. LeBrón explains the benefits and limitations of this tool:

> *Coraje* has emerged as a key target of state repression because it has the potential to create networks of solidarity grounded in a refusal of the current order—a shared understanding that this current situation should not and cannot continue. Police and local elites target expressions of *coraje* in the hopes of appealing to colonial respectability politics and norms around emotional decorum [...] These are fewer indicators of political irrationality and incivility and more an indicator of the limitations placed on Puerto Ricans' ability to challenge the colonial structures governing their everyday lives and accelerating their deaths. [3], (822–823)

The primary concern of U.S. and P.R. governance with the interests of the elite few at the expense of the livelihoods of Puerto Rican minorities is insidiously pervasive. Thus, its resistance must meticulously defend its contenders while addressing its injustices. LeBrón frames *coraje*'s strategic use as a weapon of the weak, borrowing from Scott's earlier iterations of such in his study regarding everyday practices of resistance in Malaysia [27].

Further, understanding *coraje* as a weapon of the weak uncovers colonial state violence and foments vital networks of transformative solidarity. Solidarity-based refusal, activist resistance, and mobilization tactics of feminist movement communities fuel a resiliently opposing front. I analyze how the cyclic character of celebration is a critical space under attack for Puerto Rican feminist movement communities to amplify their demands in protest and maintain the historical pertinence of protests themselves. I highlight how anti-colonial solidarity enacted through these spaces bears the lived implications of activist organizing and contentious interactions with state violence. Ultimately, I argue that the culturalization of solidarity-based networks within feminist movement communities fuels the communities' historical perseverance and the utility of weapons of the weak in the face of government and state violence.

## 2. Materials and Methods

I analyzed data collected through ethnographic participant observation as well as a small archive of digital news articles following the events observed. For my ethnographic observations, I attended the 51st International Working Women's Day march, celebrated on 8 March 2023. Coalitions of activists from different professional and political sectors who align with the pursuit and ideals of *vidas dignas* for Puerto Rican women and minorities organized the march and have spearheaded their organization in previous years. A wide array of protesters attended, including feminists, social justice activists, organizers, individual adherents, and overall supporters of justice for Puerto Rico. Police were also present at this event. The march commemorated the historical plight of women and minorities from diverse racial, disability, and age backgrounds in demanding the recognition of their rights as humans and citizens to dignified labor conditions and an existence free of targetted interpersonal or institutional violence.

I supplemented ethnographic fieldwork with content analysis of four (4) news articles and media blogs describing the event as well as sharing information about issues that emerged before, during, and after the marches. These digital blogs and news articles were archived on my computer between March and April 2023 as data was processed and analyzed. The content analyzed came from two (2) independent, feminist and Afrolatinx journals named TodasPR and Revista Étnica. Journalists from both journals were present at the march, and the journals documented and amplified the intentions and impact of the 8M march. The reach of these platforms is usually within the feminist movement communities. However, it is important to take into account that these journals also advertise and connect with a broader, virtual public through social media platforms such as Instagram, X, and Facebook. The content analyzed also came from one (1) Puerto Rican news outlet, a branch of a global news and entertainment journal under the name Metro Puerto Rico. Their content is shared through their digital website and social media platforms as well as through a weekly printed newspaper edition. Metro Puerto Rico has the greatest number of readers in the archipelago reaching 347,246 in the year 2021 compared to other local newspapers[1].

I developed codes manually for the archival data using computer writing software. I then isolated those that discussed instances of state and government interventions. As I researched strategies and mechanisms for activist organizing and state repression, I employed a deductive approach to the codes, building on previous research on state violence in Puerto Rico [28]. My data analysis focused on interactions between the feminist movement community members in their respective organizing roles and the opposing or hostile entities that were present at the march. Next, I discussed the feminist movement communities' saliency and perseverance, as well as the repressive tactics U.S. and P.R. state control deployed in reaction to the protests.

## 3. Results and Discussion

### 3.1. 8M 2023: Justicia de las Mujeres es Justicia Verde

Women's Day was originally celebrated in the United States during the early 1900s. These celebrations have commemorated the struggles, efforts, and history of women in their fight for legal rights over the years. International Working Women's Day was celebrated for the first time in Puerto Rico in the year 1972, organized by the women's front of the Puerto Rican Independence Party (PIP). In 1974, the feminist organization Mujer Intégrate Ahora (MIA) was the first to commemorate this event. Two years later, the Puerto Rican government officially declared March 8 as International Women's Day after the United Nations declared it as such in 1975. Particularly in Puerto Rico, International Women's Day is also recognized as International *Working* Women's Day to highlight the labor demands and socioeconomic contours of such protests throughout the history of its celebration. During the year 2009, a feminist coalition under the name of *Coalición 8 de marzo* (or C8M) was established. This coalition is made up of feminist and political organizations that take up the commemoration of the 8M celebratory protests every year up to date. In 2018, a legal bill was presented and approved declaring the second week of March as "Semana de la Mujer en Puerto Rico" or Puerto Rico's Women's Week. In an article published the following year by TodasPR, a feminist, independent, digital news journal in Puerto Rico, senator (and militant feminist) Ana Irma Rivera Lassén argued that extending the celebrations over the course of a week renders invisible the specific day wherein protests are held[2].

The 8M march I attended was organized by the C8M. To date, several feminist, professional, political, cultural, community, and grassroots organizations as well as other individual attendees compose the coalition. The protests took place from 4:00 to 6:00 pm in front of the Departamento de Recursos Naturales y Ambientales (DRNA) or the Department of Natural and Environmental Resources. Organized under the motto "justicia de las mujeres es justicia verde" (justice for women is green justice), this year's celebratory protest advocated for a life in harmony with natural resources, environmental, antiracist, and decolonial education, an accessible and quality health system, decent housing, the eradication of forced displacement, labor justice for women workers, energy security, and a dignified retirement[3]. These affinities were explicitly shared during a press release the week prior where speakers of the C8M announced the 8M march and stated:

> Our motto proposes a struggle from an anti-racist gender perspective that fights against a system that privileges one racial group over another; decolonial, because it advocates for breaking with the modern logic of Modernity and addressing the problems of Puerto Rico, from and for Puerto Rico; abolitionist, as it defends restorative justice against punitive justice; and anti-speciesist, since it recognizes that no species is above others[4] (*Translated*).

Further, the 8M 2023 march explicitly repudiated both local and imperial measures recently established that attack the livelihoods of working-class Puerto Ricans such as recent lawsuits against the Colegio de Profesionales del Trabajo Social de Puerto Rico (College of Professional Social Workers of Puerto Rico) from a fundamentalist sector that wants to prevent the group from advocating for the eradication of conversion therapy. Another measure the march repudiated was JJ Laura Taylor Swain's recent ruling to annul labor reform laws in Puerto Rico in compliance with the PROMESA law aforementioned[5]. The event's programming was developed and agreed upon several weeks in advance. The execution of the event was delegated to five (5) committees: logistics, claims, media, finances, and security. For my participation, I collaborated with the logistics committee to be one of the trucks to provide water to protesters and attendees.

The 8M protest observed was one of three marches held in close proximity to government offices in the capital city of Puerto Rico, San Juan. There were several dozens of people present from diverse backgrounds, age groups, races, and gender expressions with a strong presence of women and femme-presenting bodies. Some were militant ac-

tivists and feminist organizers, while others had varying degrees of connection to feminist organizations and the movement communities at large. The majority of attendees and protesters wore purple, green, yellow, or printed shirts with feminist slogans alluding to the different themes that characterized the celebratory protest. Some of the slogans read as follows, "Us against the dept", "Antipatriarchal, Feminist, Lesbian, Trans, Caribbean, Latin American", and "We stop to build a different life", as well as the slogans of other organizations that stood in solidarity with the march such as Amnesty International Puerto Rico and the Collegiate of Professional Social Workers of Puerto Rico (*translated*). Although the events took place during evening hours, the heat and humidity were palpable. Many of the protesters and attendees had brought umbrellas and sunglasses to shield from the beaming sun. The DRNA lies in front of one of the routes of the only working train on the island and a medium-sized road system that leads to different parts of San Juan. The march took place directly in front of the DRNA offices between the roads and plots of grass. Its execution kept a circular formation, called *piquete* and different signs read "I decide", "Black trans lives matter", "natural resources are not for sale", "not the land, nor the women are territory of conquest", "dignified housing provides an out for women in situations of violence", "dignified housing is a human right, not a business", "just salary and dignified retirement for all women", and "down with the patriarchy" (*translated*). The *piquete* facilitated not only the accessibility and participation of all those present, but also the same circular formation needed for the "Batey" (ceremonial center) for the participation of the 8M Barrileras, a cultural and feminist organization compromised with the Bomba dance and culture as a tool of resistance to denounce injustices, shine a light on Puerto Rican womens' struggles, express feeling, and honor revolutionary Puerto Rican women. As Bomba is a heavily gendered dance, the artivism that 8M Barrileras engages in incorporates inclusive language and celebrates cross-gender performances of the dance which actively contend with pre-established gender roles and expectations. Further, as an Afro-diasporic tradition, Bomba, in many ways, represents that commitment to antiracist and decolonial struggle in the Puerto Rican archipelago. Both the Bomba songs and the chants that filled the air of this celebratory protest reflected advocacy in favor of reproductive justice and dignified livelihoods while also critiquing the state and government for their oppressive and violent tactics. Among some of these were the following, "government, government, colonial government, they are to blame for the environmental crisis", "the penal code made me a criminal for deciding over my body and rise to protest", "if they touch one of us, they touch all of us", "our anger is collective, lets build another life" and "immigrant women, united and moving forward, immigrant families, strong and militant" (*translated*). These chants are made available digitally as well as shared physical copies among protesters at the march for everyone to be able to follow along.

Shortly after arriving, a police body of around twenty to thirty policemen and women restricted access to the event and delineated the circular formation of the protest at several points. As I brought attention to this to other committee organizers, many reflected on the continued recurrence of police obstruction under the guise of protection. The events held intercalated marching around the *piquete* while singing the different protest chants to the beat of Bomba, the public reading of the claims and complaints addressed to governmental institutions, musical interpretations of Puerto Rican feminist songs, and a SlutWalk led by an LGBTQ+ activist organization. The claims and complaints shared with the public during the public reading characterize factors that contour the forms of justice the C8M fights for including environmentalist, antiracist, decolonial, and abolitionist education, reproductive justice, accessible and good quality healthcare system, dignified housing and the elimination of forceful displacements, labor justice for working women, animal rights and climate change, electrical access and security, dignified retirement, and justice that considers people before the Puerto Rican debt. Many of these claims highlighted the role of the state and government in managing social and structural inequalities, for example:

> Every year, the mental health of people living in Puerto Rico is at greater risk due to socio-natural events and pandemics. These events have not given respite for

the recovery of our communities, especially those most vulnerable to the system. The state has not taken efficient strategic actions to address and improve the conditions in which we live after these events; on the contrary, it has contributed to making disparities more noticeable day by day[6] (*Translated*).

Here, feminists from the C8M were explicitly addressing the state for their contribution to the exacerbation of social disparities in Puerto Rico which further attacks vulnerable populations living in the islands. The occurrence of such obstructions and contributions have been documented throughout history and are brought back to living memory during the public reading.

As people trickled in throughout the event, it was noticeable that these were people who knew each other. People greeted each other with hugs and kisses on the cheek while expressing their happiness to be collectively present in this space for the purpose of defending women's and minorities' rights to dignified livelihoods. The celebratory aspect of the march was best captured through the march's dedicatory to women who have contributed to justice for Puerto Rican women among which reproductive health professional, Yari Vale and environmentalist and founder of a grassroots community organization, Tinti Deyá were highlighted. The 8M march of 2023 was not only a space to combat environmental injustice but also a space for Black Puerto Ricans to affirm their racial identities and take a stance against racist and patriarchal oppression[7]. In so doing, the C8M foregrounds their commitment to the expansion of the Puerto Rican feminist movement communities through a sense of anti-colonial solidarity that advocates for the right to a healthy life, in control of our bodies, a life that is sustainable, free, and in communion with the rest of living beings and the planet.

Instances of anti-colonial solidarity were also very apparent throughout the celebratory protest of 8M. Most notably, the security committee was mobilized on at least one occasion to make sure someone responsible for the 8M programming and execution was able to bear witness and accompany protesters directly interacting with police bodies. I was able to converse with one attendee who was taking photographs of the event and was visiting from the United States specifically to participate in the events of the 8M march. During our conversation, the young Puerto Rican woman shared she was part of those who had been displaced by the poor labor and economic conditions in Puerto Rico. She was now living in the U.S. and supporting her mother, who could not move with her and is still living in Puerto Rico. Further, this attests to the diasporic character of feminist and anti-colonial solidarity. After closing remarks were shared and the activity reached its end, many of the protesters and attendees began making arrangements to further spend time and celebrate the successful execution of another 8M march.

### 3.2. Collective Efforts behind State Violence

Many legal, political, and state organizations are invested in overshadowing the public presence of feminist activism in Puerto Rico. These structures work synchronously in the fashion of complex machines to stall and obstruct any feminist-authored progress at individual, collective, and institutionalized levels. The impact of these operations is evidenced in several ways, including the passing of legal bills that stall or dissuade feminist demands, punitive accusations, and police obstruction under the guise of protection. For example, a senator from the religious political party Proyecto Dignidad, Sen. Joanne Rodríguez Veve, denounced the 8M march to Puerto Rico's Department of Family Affairs. She refers to videos of the SlutWalk segment of the march wherein women and LGBTQ+ protesters are dancing topless in the presence of a minor and states, "I am referring to the videos so that, in carrying out their expertise and investigative faculties, the Department of Family can make the determinations it deems pertinent under the Law for the Security, Well-being, and Protection of Minors, Law 246-2011"[8]. It is worth noting her accusations are out of context. The mother of the minor was present there as well, and the partial nudity was part of a collective performance in support of bodily autonomy. Sen. Rodríguez Veve's accusations reflected tactics rooted in moral panics to discredit activists and protesters under

the guise of protecting children[9]. Further, these tactics support sociopolitical arrangements and structures grounded in the settlement of The United States's rule in the archipelago. The backbone of U.S. rule settlement in Puerto Rico is also rooted in the broader histories of exclusion and discrimination of social actors and their subjugation to the governance of The United States.

As Sen. Rivera Lassen argues, the celebratory declaration of protests that state and government officials author compromises the event's political edge as well as the safety of its attendees. Prolonging celebrations invite apolitical and acritical activities to occur independently from one another and lessen the number of attendees at each event. In so doing, the Puerto Rican government supports overshadowing and overpowering the combative character of the demands of feminist activists and the feminist social movement community at large. As the celebratory dates facilitate the public platform of any non-violent manifestation, they also invite the unsanctioned and obstructive interventions of protesters who oppose and potentially threaten the celebrations' honorees. This then authorizes the deliberate policing of protesters by government-supported metrics to dissuade political impact. This is particularly evidenced in alleged police protection through the obstruction of event routes, hostile or dismissive demeanor in most interactions, circumscribing the presence of police bodies around protests, and the baseless incarceration of protesters in the case of the marches observed and analyzed. Moreover, it was common to hear about the continuous recurrence of policing strategies alongside these annual protests in the 8M march as well as other feminist-led public events.

### 3.3. Puerto Rican Feminism as a Social Movement Community

The expansion and diversification of Puerto Rican feminist movements are accurately represented in a concept such as feminist movement communities. Several sectors of the feminist movement community mobilized on the commemorative date in different areas of the islands. Although this might result in the loss of numbers within coalitional march programming, its breadth became strategically palpable to multiple governmental structures. The feminist movement's communal character is largely responsible for the salience and perseverance of feminist activism in Puerto Rico. Activists rely on non-hierarchical, transitory, and coalitional work sustained through social media networks to organize the marches. These networks are held through freeware, cross-platform, centralized instant messaging services, are activated at the start of protest preparations, and deactivated after the event has occurred and meetings to analyze results have been held. While a social movement understands collective responses and demands, the concept of feminist movement communities aligned with ideals of gender and social justice aids in critically considering the recurrence of protests in the celebration of women, workers, and queer people in Puerto Rico. Feminist movement communities in Puerto Rico account for several actors within and beyond feminist organizations. Often, feminists who participate in other community, social justice, or political groups will mobilize efforts within these other affiliations to garner support and solidarity. Similarly, as feminist protests historically recognize the unequal treatment of women in various social institutions, many women who do not necessarily identify as feminists also fuel the feminist movement community. This community also accounts for the presence of a broader Puerto Rican community beyond insular borders as diasporic Puerto Ricans return on these dates to participate, document the events, and share their experiences locally and abroad. Some of these are often displaced Puerto Ricans who cannot financially sustain themselves in the archipelago. They use their access to U.S. citizenship to find jobs in the United States and support their elders and families who could not leave.

Puerto Rican feminist movement communities today commemorate the historical plights of women, queer people, and low-income workers at the intersections of race, migrant status, disability, and age. Many social actors find themselves vulnerable to the interlocking systems of oppression that U.S. and P.R. governance orchestrates to embody this community. The vulnerability of oppressed social locations is heightened or lessened

depending on their relation to other privileges. While unavoidable differences exist within feminist movements, the grasp of U.S. financial oversight and P.R. governmental negligence demands the cross-collaboration of all those implicated by a form of capitalist subjugation that is guided by colonial logic. Further, understanding feminist movement communities in Puerto Rico demands foregrounding anti-colonial solidarity in its management. A diverse group of protesters was appreciated throughout the march, ranging from the artivists and performers who put their art in service of resistance and protest to the cross-collaboration of political and apolitical organizations. The culturalization of solidarity-based networks and the pursuit of social justice solidifies feminist movement communities in Puerto Rico.

The celebratory protest presents a critical space for feminist movement communities to amplify the urgency of their demands in protest. This is because this commemorative date, accompanied by the calls to action that feminist activists and spokespeople issue via several private and public news outlets, bring together militant activists and organizers as well as allies, sympathizers, and others who find themselves implicated in state and government mistreatment. Additionally, many militant feminists assume the responsibility of accompanying those caught in the crossfire of protest-targeted police obstruction, as well as advocating for their safety and respect. The historical events that jumpstarted the feminist movements' insurgence and the sustained networks that resulted from them became elements of sociocultural salience. It is precisely this sociocultural salience based on anti-colonial solidarity and feminist activism that fuels the movement's survival and perseverance as a community for over four decades. Further, this sociocultural salience speaks to an alternative embodiment of Puerto Rican identity that refuses imperial subjugation and reaffirms their livelihoods as deserving of *vidas dignas*.

## 4. Conclusions

Understanding the impact of Puerto Rican feminist movements communities heavily relies on foregrounding its political as well as action-based activism. My aim is not to apolitizice a movement through a culturalizing lens; rather, I argue that understanding the motifs of a movement's saliency is partly informed by the sociocultural structures that emerge from its perseverance and sustain its continuity. There is no doubt that feminist movement communities in Puerto Rico are characterized by the ability of its members to tackle the multiple adversities that punitive government structures afford while co-constructing the blueprints of other ways of being and knowing in the world. In efforts to simultaneously eradicate state violence and bring about a new social order, feminists in Puerto Rico build communities grounded in social and environmental justice and sustain the livelihood of their social movement communities against government-authored repression and erasure. This is achieved through several efforts. First, the expansion of Puerto Rican feminist movements ensures the synchronous operation of strategic organizing and activism while also creating safe spaces for multiple communities to blossom. Second, the specialization of its members in different areas of community organizing and professional development in health, government, and education, among other social institutions, facilitates the production and redistribution of resources that are often gatekept from the communities that need them. While these efforts support the idea of strength in numbers, this alone cannot assume the protection of women and minorities against the overarching presence of state violence that the synchronous operation of the United States and Puerto Rican governments affords.

Government and state officials (local and colonial) are involved in the obstruction of feminist activism in Puerto Rico. These representatives submit bills to accuse and criminalize activists for alleged indecency. They inflame their credibility through the mobilization of moral panics relating to the safety of children. This was particularly evidenced in Senator Joanne Rodriguez Vevé's official complaint issued to Puerto Rico's Family Department within the same week of the protests that took place on March 8. This complaint related to the exposure of minors to indecent acts following the 8M march. Several parts of the state actively obstruct the operations of Puerto Rican feminist movement communities and

attempt to discredit their claims about broader societal issues that asphyxiate the lives of Puerto Rican minorities.

Celebrating the foundational achievements of Puerto Rican women, LGBTQ+ people, and workers is intrinsically implicated in protesting state violence. The commemoration of their struggles underscores their dehumanizing criminalization and evidences their strategic targeting. Future research should consider the limitations that Puerto Rican legal and government organizations implicated in the protection of these populations face. It should also investigate the differences in the lived experiences of vulnerable migrant, racialized, and/or aging communities facing detrimental levels of precarity that the U.S. and P.R. governments can afford. State violence as a global phenomenon is facing pushback from mass social mobilizations and protests often spearheaded by women and feminist activists. Puerto Rico provides an intriguing example of these interactions as we consider how the dual-fold colonial and local governments in an increasingly globalized world are present.

**Funding:** This research received no external funding.

**Institutional Review Board Statement:** The study was conducted in accordance with the Declaration of Helsinki and approved by The University of Illinois at Urbana-Champaign Office for the Protection of Research Subjects (protocol code 22730, approved 1 February 2022).

**Informed Consent Statement:** Participant consent was waived for the data set used in this article given it is ethnographic observations collected at public events. Organizers were informed of the presence of social investigators and agreed verbally to the researcher's participation.

**Data Availability Statement:** The data presented in this study are partially available on request from the corresponding author. The data are not publicly available given it is a fraction of the overall data being collected for the purpose of dissertation research that is currently ongoing.

**Conflicts of Interest:** The author is affiliated to a public state university through which the work completed for this manuscript was supported and overseen.

## Notes

[1]　"Estudio Confirma Metro es el Periódico de Mayor Lectoría en Puerto Rico", Metro Puerto Rico, 3 June 2021. Available online: https://www.metro.pr/pr/entretenimiento/2021/06/03/estudio-confirma-metro-es-el-periodico-de-mayor-lectoria-en-puerto-rico.html (accessed on 23 April 2023).

[2]　Ana Irma Rivera Lassén, "Historias del Día Internacional de las Mujeres en Tiempos de la Junta de Control Fiscal y el #8M", TodasPR, 7 March 2019. Available online: https://www.todaspr.com/historias-del-dia-internacional-de-las-mujeres-en-tiempos-de-la-junta-de-control-fiscal-y-el-8m/ (accessed on 1 March 2023).

[3]　"Coalición 8M Convoca a un Reclamo de Justicia Verde Frente al DRNA", TodasPR, 1 March 2023. Available online: https://www.todaspr.com/coalicion-8m-convoca-a-un-reclamo-de-justicia-verde-frente-al-drna/ (accessed on 1 March 2023).

[4]　A Video of the Press Release Was Shared via the C8M Facebook Page and the Press Release Letter from Which This Quote Is Taken Was Shared on the Coalition's Website. Available online: https://www.dialogosocialpr.org/en/_files/ugd/196ca8_06722b5eae8d4ad0bdb1168c70ba8f14.pdf (accessed on 1 March 2023).

[5]　No Other Information Aside from That Provided in the Press Release Video Was Found Regarding the Lawsuit against the College of Professional Social Workers of Puerto Rico. "Coalición 8M repudia anulación de la Reforma Laboral 2022", MetroPR, 4 March 2023. Available online: https://www.metro.pr/noticias/2023/03/04/coalicion-8m-repudia-anulacion-de-la-reforma-laboral-2022/ (accessed on 7 March 2023).

[6]　I Was Able to Take Notes of the Statements during Ethnographic Data Collection. Nevertheless, a Write Up of the Claims and Demands Was Shared via the C8M Website and Served to Confirm and Provide Better Accuracy to my Ethnographic Notes. Available online: https://www.dialogosocialpr.org/en/_files/ugd/196ca8_fd759839e6154ccc94e0c079fb86c5bf.pdf (accessed on 1 March 2023).

[7]　Lumarhi J. Rivera Lozada, "REPRESENTACIÓN DE LA NEGRITUD COMO ARMA EN LA LUCHA POR LAS MUJERES", Revista Étnica, 26 March 2023. Available online: https://www.revistaetnica.com/blogs/news/representacion-de-la-negritud-como-arma-en-la-lucha-por-las-mujeres (accessed on 23 April 2023).

[8]　"Rodríguez Veve pide a Familia que Investigue Participación de Niña en Manifestación del Día de la Mujer", Metro Puerto Rico, 13 March 2023. Available online: https://www.metro.pr/noticias/2023/03/13/rodriguez-veve-pide-a-familia-que-investigue-participacion-de-nina-en-manifestacion-del-dia-de-la-mujer/ (accessed on 13 March 2023).

9      No further information regarding this accusation was found during the process of archiving news articles.

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
