# Peer review of "“Festival y Protesta”: The Integral Role of Protesting State Violence in Celebrating Puerto Rican Women and Feminists"

_societies, doi:10.3390/soc13120251_

Round 1

Reviewer 1 Report

Comments and Suggestions for Authors

Thank you for asking me to review this submission, from which I learned a lot about the state of play regarding feminist, LGBTQ+ and other marginalised communities in Puerto Rico. I enjoyed reading your work. I recommend publication with only a few minor suggestions, which I think would further bolster the scholarly contribution of your work.

In the first section, Puerto Rico: State and Government, perhaps a clearer account of the ways in which the socio-economic grip of the USA (especially the Financial Oversight and Management Board) over Puerto Rican marginalised communities would inject this essay with more urgency, and would clarify how Puerto Rican SMCs diverge from all sorts of US mainstream feminisms.

In the 3rd section, Results and Discussion (especially 3.2 on state violence), some short commentary on the ways in which policing is organised in Puerto Rico might be useful. I would suggest you check out Verso Books on policing –  https://www.versobooks.com/en-gb/collections/police-prisons?_pos=1&_psq=police&_ss=e&_v=1.0  

Comments on the Quality of English Language

Finally, I would suggest you go over the submission as there are a very few minor issues with grammar.

Author Response

Thank you for taking the time to review my manuscript and for your thoughtful feedback. The following revisions have been made and highlighted on the latest manuscript submission:

Comment 1: State and Government, perhaps a clearer account of the ways in which the socio-economic grip of the USA (especially the Financial Oversight and Management Board) over Puerto Rican marginalised communities would inject this essay with more urgency, and would clarify how Puerto Rican SMCs diverge from all sorts of US mainstream feminisms.

Response 1: I agree, accordingly, section 1.1 (lines 82-97) expand on the ways in which  the socio-economic grip of the USA over marginalized communities in Puerto Rico. The added language shines light on the uniqueness of the situation that demands cross-organizational and cross-political involvement for all marginalized Puerto Ricans implicated. Puerto Rican Feminism does learn from and build upon US feminisms which is important to its development. 

Comment 2: some short commentary on the ways in which policing is organised in Puerto Rico might be useful. 

Response 2: I agree, accordingly, section 3.2 (lines 340-347) expand on the ways in which policing is organized and enacted in Puerto Rico

I hope to have accurately implemented your recommendations and am willing to continue to implement any further feedback. Thank you.

Reviewer 2 Report

Comments and Suggestions for Authors

The results section would be more descriptive and informative if quotes from protestors or others involved in the ethnographic research were included. It seems that conclusion were drawn based on little descriptive evidence presented to the reader. Ethnographic research gives the researcher the chance to provide in-depth and rich data in the form of examples and personal descriptions of events studied. I fell that this is lacking in this paper. I think that the findings would be enhanced with inclusion of specific examples and possibly quotes from participants in protests, and so on.

Author Response

Thank you for taking the time to review my manuscript and for your thoughtful feedback. The following revisions have been made and highlighted on the latest manuscript submission:

Comment 1: "... I think that the findings would be enhanced with inclusion of specific examples and possibly quotes from participants in protests, and so on."

Response 1: I agree, accordingly, sections 3.1 (lines 281-284, 304-315) and 3.2 (351-356, 361-363) have added descriptive language and specific examples that relate to the overall results and conclusions of the study. 

I hope to have accurately implemented your recommendations and am willing to continue to implement any further feedback. Thank you.